# In Vitro Analysis of Biological Activity of Circulating Cell-Free DNA Isolated from Blood Plasma of Schizophrenic Patients and Healthy Controls

**DOI:** 10.3390/genes13030551

**Published:** 2022-03-20

**Authors:** Elizaveta S. Ershova, Galina V. Shmarina, Lev N. Porokhovnik, Natalia V. Zakharova, George P. Kostyuk, Pavel E. Umriukhin, Sergey I. Kutsev, Vasilina A. Sergeeva, Natalia N. Veiko, Svetlana V. Kostyuk

**Affiliations:** 1Molecular Biology Laboratory, Research Centre for Medical Genetics, 115522 Moscow, Russia; es-ershova@rambler.ru (E.S.E.); sakmarariver@yahoo.com (G.V.S.); pavelum@mail.ru (P.E.U.); kutsev@mail.ru (S.I.K.); tracytheplane@gmail.com (V.A.S.); satelit32006@yandex.ru (N.N.V.); svet-vk@yandex.ru (S.V.K.); 2N.A. Alekseev Clinical Psychiatric Hospital No. 1, 117152 Moscow, Russia; nataliza80@gmail.com (N.V.Z.); kgr@yandex.ru (G.P.K.); 3Department of Physiology, I.M. Sechenov First Moscow State Medical University, 119991 Moscow, Russia

**Keywords:** schizophrenia, cell-free DNA, TLR9, AIM2, STING, RIG-I, EEA1, HMGB1

## Abstract

Schizophrenia is associated with low-grade systemic inflammation. Circulating cell-free DNA (c-cfDNA) belongs to the DAMP class. The major research question was: can the c-cfDNA of schizophrenic patients (sz-cfDNA) stimulate the DNA sensor genes, which control the innate immunity? We investigated the in vitro response of ten human skin fibroblast (HSF) lines to five DNA probes containing different amounts of a GC-rich marker (the ribosomal repeat) and a DNA oxidation marker (8-oxodG) including sz-cfDNA and healthy control c-cfDNA (hc-cfDNA) probes. After 1 h, 3 h, and 24 h of incubation, the expression of 6 protein genes responsible for cfDNA transport into the cell (EEA1 and HMGB1) and the recognition of cytosolic DNA (TLR9, AIM2, STING and RIG-I) was analyzed at the transcriptional (RT-qPCR) and protein level (flow cytometry and fluorescence microscopy). Additionally, we analyzed changes in the RNA amount of 32 genes (RT-qPCR), which had been previously associated with different cellular responses to cell-free DNA with different characteristics. Adding sz-cfDNA and hc-cfDNA to the HSF medium in equal amounts (50 ng/mL) blocked endocytosis and stimulated *TLR9* and *STING* gene expression while blocking *RIG-I* and *AIM2* expression. Sz-cfDNA and hc-cfDNA, compared to gDNA, demonstrated much stronger stimulated transcription of genes that control cell proliferation, cytokine synthesis, apoptosis, autophagy, and mitochondrial biogenesis. No significant difference was observed in the response of the cells to sz-cfDNA and hc-cfDNA. Sz-cfDNA and hc-cfDNA showed similarly high biological activity towards HSFs, stimulating the gene activity of TLR9 and STING DNA sensor proteins and blocking the activity of the AIM2 protein gene. Since the sz-cfDNA content in the patients’ blood is several times higher than the hc-cfDNA content, sz-cfDNA may upregulate pro-inflammatory cytokines in schizophrenia.

## 1. Introduction

Inflammation is believed to play an essential role in the development and persistence of psychosis and cognitive impairment in schizophrenia [1,2,3]. An acute psychotic episode is associated with low-grade systemic inflammation in some patients. Several biomarker studies have found associations between pro-inflammatory cytokines and schizophrenia [4,5]. The authors point to a variety of systemic inflammation reasons in schizophrenia. Among other things, it has been suggested that circulating cell-free DNA (c-cfDNA) may play a role in systemic inflammation maintenance in a schizophrenic patient [6,7].

C-cfDNA is widely used in clinical practice to confirm the diagnosis and prognosis of the disease and monitor the patient’s response to therapy. The main sources of circulating cell-free DNA are apoptosis, necrosis and active secretion [8]. The pool of c-cfDNA fragments can contain the DNA fragments of a tumor, fetus or transplanted organ, allowing fluid biopsy without traumatic intervention. The c-cfDNA concentration was found to increase in patients with sepsis, cancer, cardiovascular and autoimmune diseases [9,10,11,12,13,14,15]. Pregnancy, physical exercise and psychosocial stress also stimulate an increase in c-cfDNA concentration [16,17,18,19,20]. It was shown that the c-cfDNA level is considerably increased in the case of some CNS diseases, such as focal epilepsy [21], multiple sclerosis [22], stroke [23,24] or brain tumor [25,26].

Recently, c-cfDNA has been considered not only as a diagnostic marker but also a potential therapeutic target [27,28,29,30]. In vitro experiments have shown that the c-cfDNA of patients with cardiovascular and autoimmune diseases has a pronounced biological activity with respect to various types of human cells [9,31,32,33]. This is due to c-cfDNA’s molecular characteristics, which significantly distinguish c-cfDNA from the genomic DNA (gDNA) that functions inside the cell.

GC-rich genome fragments (gc-DNA) are accumulated in the c-cfDNA pool. C-cfDNA isolated from healthy subjects contains approximately 54% of GC pairs [34] compared to 42% in gDNA [35]. The content of such gc-DNA as mitochondrial DNA (mtDNA) and the ribosomal repeat (rDNA) in c-cfDNA significantly increases in pathology or external influences, which are accompanied by chronic oxidative stress and upregulated cell death [36,37,38,39].

C-cfDNA differs from gDNA by its high oxidative modification level. The content of the oxidation marker 8-oxodG in c-cfDNA is several times higher than in gDNA [6]. The oxidation of c-cfDNA fragments significantly increases their biological activity [40,41,42,43,44]. Oxidized cfDNA fragments (oxy-DNA) stimulate ROS synthesis in cells via boosting NADPH oxidase 4 (NOX4) expression. Oxy-DNA is a signaling molecule involved in adaptive responses and the bystander radiation effect. Fragments of oxy-DNA, on the one hand, can induce instability of the genome in tumor cells, and, on the other hand, they increase cell survival by blocking apoptosis, affecting tumor therapy. Oxy-DNA with low 8-oxodG content stimulates nitric oxide synthesis by endothelial NO synthase (eNOS) in endothelial cells. A high cfDNA oxidation level is accompanied by an eNOS activity block [45,46].

The oxidation of cfDNA significantly improves its ability to penetrate the cell across the cell membrane [47]. Oxidized cfDNA fragments penetrating the cytoplasm can potentially induce different responses, including the stimulation of signaling pathways associated with cytoplasmic nucleic acids receptors, such as Toll-like receptor 9 (TLR9), absent in melanoma 2 (AIM2), cyclic GMP-AMP synthase (cGAS), and retinoic acid-inducible gene-1-like receptors (RIG-I).

The changes in c-cfDNA properties in schizophrenia have also been studied. The total c-cfDNA concentration in the blood of patients from China and Russia was investigated [6,7,48,49]. All the studies reported a two- to three-fold c-cfDNA increase in SZ patients. The concentrations of mtDNA [38], rDNA [7], GC-rich Alu repeats [50], and oxy-DNA [6] in the blood plasma of SZ patients exceeded the concentrations in the blood plasma of healthy controls (hc-cfDNA).

The increased plasma levels of gc-DNA and oxy-DNA suggest that sz-cfDNA is a biologically active molecule. For example, the authors [50] found a positive relationship between the contents of Alu and interleukin-1β and interleukin-18 in SZ patients. However, a priori, we cannot determine what could be responsible for the potential biological activity of sz-cfDNA: a high concentration of c-cfDNA fragments or an altered composition of sz-cfDNA compared to hc-cfDNA fragments. The literature contains sporadic reports on the biological activity of hc-cfDNA samples isolated from the plasma of healthy subjects [32,46]. At the same time, hc-cfDNA contains more biologically active gc-DNA and oxy-DNA fragments than cellular gDNA [6,7,34].

Endocytosis is one of the possible ways for endogenous cfDNA penetration into the cells. Early endosome antigen 1 is a membrane-bound Rab5 effector protein specific to the early endosome. EEA1 plays an important role in membrane trafficking. The content of this protein increases in response to endocytosis process activation in the cell pool [51,52].

One of the modes for cfDNA to penetrate into the cells is the interaction of CpG-rich fragments with TLR9 receptors. TLR9s are important human innate immunity receptors that participate in several cell functions, including immune responses. TLR9 recognizes the unmethylated CpG motif. TLR9 is expressed both on the cell surface and in endosomes. The formation of surface TLR9 complexes with GC-DNA is one of the mechanisms for gc-DNA penetration into the cell. TLR9 activation induces a MyD88-dependent downstream signaling pathway to upregulate IRF3-based type 1 IFN production and NF-κB mediated pro-inflammatory cytokine generation [53,54].

AIM2 is a component of the innate immune system that functions as a cytoplasmic dsDNA sensor involving self DNA. Together with the adapter ASC protein, AIM2 forms a caspase-1 activating complex known as the AIM2 inflammasome. It is assumed that aberrant activation of AIM2 from self-DNA is a driver of inflammation [54,55]. It was also shown that defective AIM2 inflammasome signaling results in decreased neural cell death, both in response to DNA damage-inducing agents and during neurodevelopment [56].

The cytosolic DNA sensor cyclic guanosine monophosphate-adenosine monophosphate synthase (cGAS) detects cytosolic DNA and then mediates downstream responses through the STING-DNA-sensing pathway and downstream interferon signaling. STING activation is also involved in NF-kB, MAPKs, and STAT6 activation and in stimulating autophagosome formation. Thus, cGAS-STING signaling plays a crucial role in inflammation [54,57,58].

RIG-I functions as a cytosolic receptor detecting exogenous RNAs. RIG-I is able to detect 5′-triphosphorylated dsRNA transcribed from AT-rich dsDNA by DNA-dependent RNA polymerase III (Pol III) [42,59,60,61]. Apparently, cfDNA penetrating into the cell is transcribed with the formation of dsRNA molecules that interact with RIG-I.

HMGB1 has multiple functions in cells. This protein functions both in the cell nucleus and in the extracellular environment. HMGB1 can bind negatively charged endogenous extracellular DNA to facilitate its cellular uptake via RAGE-receptor-mediated endocytosis [62,63,64]. HMGB1 enhances the cell response to DNA fragments that induce TLR9 and cGAS-STING signaling pathways.

The major objective of our study was to compare the biological activity of the c-cfDNA samples isolated from the blood plasma of healthy controls and SZ patients. We used human skin fibroblasts (HSFs) cultured in a serum-free medium as an in vitro model. The sz-cfDNA or hc-cfDNA samples were added to the culture medium at the same concentration. Previously, we showed that this model was able to distinguish the cell responses to gDNA and oxy-DNA fragments in the culture medium [65]. In the present study, we analyzed the changes in genes expression of nucleic acid sensor proteins (TLR9, AIM2, STING, RIG-I) and proteins involved in the biopolymers’ transfection into cells (EEA1 and HMGB1) at the transcriptional and translational levels in human cells in response to cfDNA fragments with different properties, including sz-cfDNA, hc-cfDNA and the model gDNA, oxy-DNA and gc-DNA.

## 2. Results

### 2.1. DNA Probes

Adding gDNA to the medium simulated an increase in cfDNA concentration without significant changes in the GC composition and the oxidation level.

Probe oxy-gDNA simulated an alteration of the cfDNA oxidation fraction while the total cfDNA concentration increased. Probe gc-DNA simulated an increase in the concentration of GC-rich repeats.

Oxy-gDNA, sz-cfDNA, and hc-cfDNA appeared to contain fragments of various lengths, including a small fraction of short fragments approximately 200 pairs long (~5% of the total DNA) (Figure 1A).

### 2.2. Interaction of the DNA Probes with HSFs

The FCA assay could show whether DNA fragments penetrated into the cells since the procedure of the removal of cells from the carrier includes treatment with EDTA and trypsin solutions. In contrast to microscopic data, FCA showed weak DNA probe penetration into the cells of each HSF line. We observed the maximum FL-signal increase in the presence of a sz-cfDNA sample in sz-HSF1 cells (Figure 1C(2)).

### 2.3. Early Endosome Antigen 1 (EEA1) Expression

We found no significant differences in the RNA *EEA1* level in sz-HSFs and hc-HSFs (*p* > 0.05). The addition of sz-cfDNA and hc-cfDNA samples to the cell culture medium was accompanied by a significant increase in the amount of RNA *EEA1* (Figure 1A). The rest of the DNA probes did not affect the RNA*EEA1* level.

EEA1 protein levels in the cells were analyzed using FCA (Figure 2B–E). Figure 2B shows an example of EEA1 analysis in sz-HSF4 cells after 24 h of cultivation. Adding DNA probes to the culture medium led to a decrease in the EEA1 level in all the cells of the population.

Figure 2C shows mean EEA1 levels in HSFs. Primary data are presented as median signal values minus median signal background values (ΔEEA1). The index EEA1 was introduced. To calculate the EEA1 index, ΔEEA1 for each HSF was divided by the maximum ΔEEA1 in the sample (n = 180). A comparison EEA1 of each of the HSFs with the corresponding control is shown in Figure 2D. Sz-cfDNA or/and hc-cfDNA incubated with cells for more than 1 h caused a decrease in the amount of EEA1 in HSFs. The maximum decrease in the level of EEA1 in sz-HSFs was observed in the presence of the sz-cfDNA probe after 3 h of cultivation.

Therefore, we have found that in the presence of sz-cfDNA and hc-cfDNA, the amount of EEA1 protein in the cells decreases with an increase in the RNA *EEA1* level. Probably, the decrease in the *EEA1* gene product amount in response to these DNA probes occurs at the translation level and/or due to sEEA1 protein degradation.

A decrease in the level of EEA1 in the cells exposed to DNA probes indicates a blockage of endocytosis. The sz-cfDNA and hc-cfDNA samples have the maximum blocking effect (Figure 2E). EEA1 levels changes induced by oxy-DNA, sz-cfDNA, and hc-cfDNA positively correlate with each other (*p* < 0.001). These probes differ from others by a higher oxidation level (see Table 3 in Section 4). Thus, there seems to be a protective molecular mechanism in the cell, which blocks endocytosis in response to the oxidized cfDNA penetration into the cytoplasm.

Nevertheless, we have repeatedly noted that cfDNA oxidation [66] promotes effective cfDNA penetration into the cells. The penetration mechanism of oxidized DNA fragments through the membrane is not yet understood.

### 2.4. Toll-like Receptor 9 (TLR9) Expression

The amounts of RNA *TLR9* in sz-HSFs and hc-HSFs did not differ significantly (*p* > 0.05). One h after sz-cfDNA and hc-cfDNA addition, we observed a significant increase in the RNA *TLR9* amount. After 3 h, no differences in RNA *TLR9* levels were observed for DNA probes; however, after 24 h of culturing in the presence of sz-cfDNA, RNA *TLR9* levels increased in all the HSF populations (Figure 3A). Thus, sz-cfDNA and hc-cfDNA probes are strong stimulators of *TLR9* gene transcription.

Data on TLR9 protein content in sz-HSFs and hc-HSFs are shown in Figure 3B–E. Figure 3B provides an example of TLR9 analysis in hc-HSF4 (1 h). The gcDNA and sz-cfDNA probes stimulate an increase in the level of TLR9 protein in all the cells. In this case, a subpopulation of cells appears (~10% of the total population) with a very high TLR9 expression level.

Figure 3C presents mean FL-TLR9 values normalized to the maximum value in the sample. TLR9 protein expression in the control sz-HSFs and hc-HSFs populations depends on the cultivation duration. After 24 h, the amount of TLR9 protein in the control cells increases significantly (*p* < 0.05). The TLR9 level in sz-HSFs and hc-HSFs increased, while the RNA *TLR9* level slightly decreased. This fact may indicate a more efficient translation process or the protein degradation rate decrease. Earlier, we found a similar effect for MCF7 cells [41].

A comparison of each of the HSFs in the presence of the DNA probes with the corresponding control is shown in Figure 3D. An unambiguous conclusion about TLR9 level increase in response to the different DNA probes cannot be made based on this study. This is due to the individual HSFs lines’ sensitivity to the various DNA probes. DNA probes stimulate an increase in the TLR9 amount at different times in three populations of sz-HSFs (1, 3 and 5). In the presence of the sz-cfDNA probe, TLR9 increases in all hc-HSFs after 1 h or 3 h (Figure 3D). The changes in the TLR9 levels in the presence of various DNA probes correlate negatively with baseline TLR9 levels in the control cells (*p* < 0.01). If the control cells contain a low TLR9 level, then DNA probes stimulate a significant increase in the protein amount. In the cells with high TLR9 levels, the protein content is reduced in the presence of DNA probes. Thus, cfDNA fragments have a modulating effect on the TLR9 protein level in HSFs.

Changes in the TLR9 level in ten HSFs upon the addition of DNA probes (1 h) correlate with each other. We found a highly positive correlation in the TLR9 level change for gc-DNA, sz-cfDNA and hc-cfDNA probes (*p* < 0.001). These probes had the highest abundance of ribosomal fragments containing CpG motifs (see Table 3 in Section 4). In other cases, the correlation was less pronounced (*p* < 0.04).

To show that cfDNA fragments bind to TLR9, we used FITC-labeled DNA probes (Figure 3F). TLR9 was visualized with PE-labeled antibodies. In the presence of all labeled DNA probes, we observed approximately the same pattern of DNA (green) and TLR9 (red) distribution. Some of the signals coincided, but about half of the signals from the labeled DNA did not coincide with the signals from TLR9.

### 2.5. Absent in Melanoma 2 (AIM2) Expression

The amounts of RNA *AIM2* in sz-HSFs and hc-HSFs did not differ significantly (*p* > 0.05). Changes in the cfDNA properties significantly block *AIM2* gene expression. RNA*AIM2* levels were maximally reduced in the presence of sz-cfDNA and hc-cfDNA probes (Figure 4A).

In 8 of 10 lines studied, the amount of AIM2 protein decreases when the time of cultivation increases (Figure 4C). In two strains (sz-HSF4 and hc-HSF1) with an initially low protein level, the AIM2 amount increases with cultivation, but then (after 48–72 h) it also starts to decrease. Previously, we described a similar dependence of the amount of AIM2 on the cultivation time for MCF7 cells [67].

The amount of AIM2 protein in the HSFs is generally decreased in the presence of the DNA probes (Figure 4B–G). When the cfDNA properties change, the AIM2 protein amount decreases in almost all cells of the population (example in Figure 4B for the hc-HSF5 (1 h)). The exposure of HSFs to DNA probes for 24 h induces a decrease in AIM2 levels in all studied lines (Figure 4C,D). After 24 h, there was a significant correlation in the decrease in AIM2 levels in the cells in the presence of gcDNA, sz-cfDNA and hc-cfDNA probes (*p* < 0.01).

Figure 4F shows the ratio of *TLR9* and *AIM2* RNA levels and TLR9 and AIM2 proteins in the cells. The sz-cfDNA probe caused the maximum changes in the RNA*TLR9*/RNA*AIM2* and TLR9/AIM2 ratios. Changes in the TLR9/AIM2 ratio in the presence of sz-cfDNA and hc-cfDNA probes in 10 HSFs positively correlated with each other after 3 h (Rs = 0.95; *p* << 0.0001) and 24 h (Rs = 0.67; *p* = 0.035).

### 2.6. Stimulator of Interferon Genes (STING) Expression

The *TMEM173* gene encodes the STING protein. Changes in the cfDNA profile in the cultivation medium stimulate *TMEM173* gene expression. Sz-cfDNA and hc-cfDNA samples 1 h after addition to the medium stimulated a 2–40 fold increase in the RNA*MEM173* amount in HSFs (Figure 5A). The gc-DNA probe also stimulated *TMEM173* expression, but at a later time. After 24 h, the maximum level of RNA *TMEM173* was observed in the presence of gc-DNA and sz-cfDNA probes. Oxy-DNA and gDNA had little effect on RNA *TMEM173* levels in HSFs.

The amount of STING protein in HSFs in the presence of DNA probes is increased compared to the control (Figure 5B–E). Sz-HSFs and hc-HSFs do not significantly differ by STING levels (*p* > 0.05). Maximum STING levels are detected after 24 h of incubation. The amount of protein increases in all cells of the population (example distribution in Figure 5B for hc-HSF4 (24 h)). Gc-DNA, sz-cfDNA and hc-cfDNA probes stimulate the maximum STING level increase in the cells (Figure 5C,D). STING protein level changes in the presence of gcDNA probe (1 h) were positively correlated with changes induced by sz-cfDNA probe in 1 h (Rs = 0.82; *p* = 0.004).

### 2.7. Retinoic Acid-Inducible Gene I (RIG-I) Expression

The RIG-I protein is encoded by the *DDX58* gene. DNA probes stimulated the maximum increase in the RNA*DDX58* amount 1 h after adding to the medium. Gc-DNA, sz-cfDNA and oxy-DNA probes have the greatest effect (Figure 6A).

A typical example of cell distribution by RIG-I protein level is shown in Figure 6B. Protein levels increase or decrease under the influence of the DNA probes in all cells of the population. Mean values of protein levels in the cells and comparison with controls are shown in Figure 6C,D. The summary analysis of the changes in RIG-I protein level upon changes in the cfDNA properties does not allow an unambiguous conclusion to be made about increases in the RIG-I level in the cells. This is due to the individual HSF lines’ sensitivity to the various DNA probes.

In five HSFs (sz-HSF (2,4) and hc-HSF (1,3,5)), the RIG-I protein content increased maximally 1 and 3 h after the addition of the DNA samples. The maximum increase was recorded for gc-DNA, sz-cfDNA and oxy-DNA samples. In other HSFs, the protein content, as a rule, decreased in the presence of DNA probes.

Changes in the RIG-I protein expression levels in the ten HSF lines exposed to all DNA probes showed a positive correlation with each other. The maximum correlation after 1 or 3 h was found for gc-DNA, hc-cfDNA and sz-cfDNA (*p* < 0.001).

### 2.8. High Mobility Group Box 1 (HMGB1) Expression

The RNA*HMGB1* levels in HSFs increased in the presence of oxy-DNA, gc-DNA and sz-cfDNA probes after 1 h of exposure to DNA probes (Figure 7A).

Data on HMGB1 protein content in HSF populations are shown in Figure 7B–E. Figure 7B shows an example of changes in HMGB1 levels in one of the lines (sz-HSF3 (1h)). The response of cells to the DNA probes is individual. Different lines of HSFs respond differently to changes in the properties of the cfDNA environment. Notably, 1 and 3 h after the addition of DNA probes, the average HMGB1 level increases only in sz-HSFs (1,5) and hc-HSFs (3,5). After 24 h, a significant increase in the HMGB1 level was observed for hc-HSF1 in the presence of all the DNA probes. For other HSFs, the protein content decreased or did not increase significantly.

Changes in the presence of sz-cfDNA and hc-cfDNA probes significantly correlate with the changes observed in the presence of a gc-DNA probe at all cultivation periods and in the presence of gDNA and oxy-DNA probes during long-term cultivation (*p* < 0.001).

Figure 7F provides an example of the HMGB1 protein localization analysis in sz-HSF1 cells that were incubated for 1 h in the presence of sz-cfDNA. In the presence of DNA probes, signals from HMGB1 appear in the cytoplasm (green), which coincide with DNA signals (red).

### 2.9. Correlation Analysis of Changes in the Protein Levels in HSFs

All the data analysis performed for 5 DNA probes (10 HSFs, 6 proteins, 3 time intervals, n = 180) showed a positive correlation between the DNA probes and the level of protein changes in the cells. The maximum correlation was observed for 3 DNA probes: gc-DNA, sz-cfDNA and hc-cfDNA (Rs = 0.70–0.75, *p* << 0.00001, n = 180).

Then, we analyzed how synchronously the levels of six analyzed proteins change in ten HSFs in the presence of DNA probes in the culture medium. A correlation analysis of all the data showed a significant positive correlation between the changes in the levels of three proteins: AIM2, RIG-I and HMGB1. A decrease in the AIM2 level in the cells relative to the control in the presence of DNA probes is associated with a relative decrease in the RIG-I and HMGB1 levels. The maximum correlation was observed for the AIM2—RIG-I pair (Rs = 0.35, *p* << 0.00001). There is also a positive correlation between changes in TLR9 and HMGB1 (Rs = 0.18, *p* = 0.027) and a negative correlation between TLR9 and RIG-I (Rs = −0.17, *p* = 0.038). Changes in AIM2 correlate negatively with changes in STING (Rs = −0.18, *p* = 0.03). Changes in EEA1 correlate positively with changes in RIG-I (Rs = 0.19, *p* = 0.018).

The results of a more detailed analysis are presented in Table 1. When gDNA, gc-DNA, sz-cfDNA and hc-cfDNA samples were added to the medium for 1 h, we observed a negative correlation between changes in TLR9 and RIG-I protein levels. The more theTLR9 level increased in the presence of DNA probes, the more the RIG-I protein level decreased. Comparison of the data shown in Figure 3D and Figure 6D shows that an increase in TLR9 level in HSFs (1 sz, 3 sz, 5 sz, 2 hc, 4 hc) in the presence of gc-DNA, sz-cfDNA and hc-cfDNA probes after 1 h is associated with a decrease in RIG-I protein levels in the same HSFs lines.

After 3 h of exposure to gDNA, gc-DNA, sz-cfDNA and hc-cfDNA probes, there was a positive correlation between changes in TLR9 and HMGB1. After 24 h of exposure to oxy-DNA and sz-cfDNA probes, changes in TLR9 correlate positively with changes in the STING level.

For gc-DNA, sz-cfDNA and hc-cfDNA probes, after 1 and/or 3 h, a positive correlation was observed between changes in AIM2 and RIG-I protein levels. For gDNA, oxy-DNA, gc-DNA and sz-cfDNA samples, there was a positive correlation between AIM2 and HMGB1 changes after 3 h or 24 h. In the case of oxy-DNA (1 h), changes in AIM2 were negatively correlated with STING changes. In the presence of gDNA, sz-cfDNA and hc-cfDNA (1 h), there is a positive correlation for RIG-I and EEA1.

### 2.10. Changes in the Some Genes’ RNA Level

Table 2 summarizes average changes of mRNA levels for 33 genes in 10 HSF lines, which, according to our previous studies, increase the transcriptional activity in response to various DNA fragments in the culture medium of different cell types.

The set of genes included genes for transcription factors that regulate the processes of cell proliferation and survival (*NFKB1*, *STAT3*, *STAT6*, *PPARG2*, *TB53*); genes for the mitogen-activated protein kinase family (*MAPK1*, *MAPK3*) and genes for the protein kinases *mTOR* and *AKT2*; genes for proteins that regulate the processes of repair (*ATM*, *ATR*), autophagy (*ATG16L1*, *BECN1*) and apoptosis (*APAF1*, *AIFM1*, *BAX*, *BAK1*, *BIRC2*, *BIRC3*, *BCL2*, *BCL2A1*, *BCL2L1*); genes for cell cycle proteins (*CCND1*, *CDKN2A*, *CDKN1A*), adhesion proteins (*ICAM*, *VCAM*, *SELE*) and cytokines (*IL-8*, *IL1B*); genes regulating mitochondrial biogenesis (*MFN1*, *FIS1*) and a gene for a protein that hydrolyzes GC-rich DNA in the cytoplasm (*ENDOG*). Most of the listed genes are more or less associated with signaling pathways that include signal transmission from DNA sensors to the cell nucleus.

We found an increase in the amount of RNA of most genes after 1 h of exposure to sz-cfDNA and hc-cfDNA. The only exception was the *ICAM* adhesion protein gene. For the rest of the DNA samples, the response of the cells was less pronounced. The maximum differences in the HSF responses to the DNA probes within 1 h were observed for the genes *mTOR*, *AKT2*, *TB53*, *ATM*, *AIFM1*, *BAK1*, *APAF1*, *BCL2*, *CCND1*, *SELE*, *MFN1*, *ENDOG*, *ATG16L1* and *IL1B*. Changes in the amount of RNA of 33 genes in the presence of sz-cfDNA and hc-cfDNA after 1 h positively correlate with each other (Rs = 0.78; *p* << 0.0001; n = 33). A positive correlation is observed for sz-cfDNA and gc-DNA (Rs = 0.42; *p* = 0.016). Changes in the level of RNA genes in the presence of gDNA positively correlate with changes in the presence of gc-DNA (Rs = 0.80; *p* << 0.0001).

After 3 h, the general picture of the HSF response to cfDNA changes. The magnitude of the effects is significantly reduced for a number of genes. After 3 h of cultivation, changes in the RNA amount of 33 genes in the presence of sz-cfDNA and hc-cfDNA still positively correlate with each other (Rs = 0.47; *p* = 0.006). There is also a high correlation between the gc-DNA and gDNA probes (Rs = 0.79; *p* << 0.0001).

After 24 h of cell exposure to gc-DNA, sz-cfDNA and hc-cfDNA, a secondary rise in the RNA levels of several genes is observed. The strongest response was observed in the presence of sz-cfDNA and hc-cfDNA probes. 24 h after the addition of the DNA probes to the cells, a positive correlation remains in the magnitude of the amount of RNA of the 33 genes changes between sz-cfDNA and hc-cfDNA (Rs = 0.62; *p* = 0.0001).

Compared to the sz-cfDNA probe, hc-cfDNA, after 24 h, induces higher RNA levels of the *STAT3*, *MTOR*, *TP53*, *ATM*, *AIFM1*, *BAX*, *BIRC2*, *BIRC3*, *APAF1*, *BCL2*, *BCL2A1*, *BCL2L1*, *CCND1*, *VCAM*, and *IL1B* genes. The gc-DNA probe, in comparison with sz-cfDNA after 24 h, stimulates higher RNA levels of the *PPARG2*, *MAPK3*, *BAX*, *BCL2A1*, *CDKN1A*, *ICAM*, *VCAM*, and *IL1B* genes. Oxy-DNA stimulates higher levels of RNA of the *BAX*, *BCL2L1*, *CDKN1A*, *ICAM*, and *IL1B* genes. The gDNA probe induces higher RNA levels of the *PPARG2*, *BAX*, *CCND1*, *CDKN1A*, *VCAM*, and *VCAM* genes.

## 3. Discussion

Circulating cfDsNA belongs to the damage/danger-associated molecular pattern (DAMP) family. DNA fragments of dead cells circulating in the extracellular environment are able to penetrate into the cytoplasm and interact with nucleic acid sensors that recognize the DNA of viruses and bacteria for a subsequent immune response. Based on the previously studied properties of sz-cfDNA [6,7], we hypothesized that sz-cfDNA can enter the cells and stimulate signaling pathways associated with the nucleic acid sensors. Activation of these pathways can lead to a systemic inflammation that is typical for schizophrenia. To test this hypothesis, we compared the biological activity of sz-cfDNA and hc-cfDNA samples in vitro on a model system—subconfluent HSFs cultured in a serum-free medium. We have previously shown that this model allows the biological activity of different DNA probes added to the culture medium to be compared [41].

The cfDNA probes interact with the cell surface. However, only a small part of cfDNA fragments penetrate into the cells (Figure 1). We found that altering the cfDNA characteristics was accompanied by a decrease in the endocytosis level. The sz-cfDNA and hc-cfDNA have the maximum blocking effect (Figure 2).

Despite the endocytosis block, some of the cfDNA fragments undoubtedly penetrate into the HSF cytoplasm (Figure 1). The appearance of cfDNA fragments in the cytoplasm stimulates another defense mechanism—an increase in the *ENDOG* gene activity. ENDOG endonuclease hydrolyzes GC-rich DNA fragments, and the sz-cfDNA and hc-cfDNA probes have the maximum stimulating effect on *ENDOG* (Table 2). The penetration of cfDNA fragments into the cell is also indicated by a change in the activity of the genes of DNA sensors and proteins involved in transmitting signals from DNA sensors to the nucleus.s

Microscopic data showed that some of the cfDNA fragments colocalize with TLR9 (Figure 3F). In response to the change in the cfDNA concentration, the amount of protein TLR9 increases in cells with an initially low TLR9 expression level. In the cells with high protein contents, the amount of TLR9 decreases in the presence of DNA probes. We found a positive correlation in the TLR9 level change for gc-DNA, sz-cfDNA and hc-cfDNA. These probes contain the largest percentage of CpG motifs and stimulate the TLR9-NF-kB-cytokine signaling pathway more strongly. This is indicated by the highest levels of RNA*NFKB1* and RNA*IL-8* in the cells cultured in the presence of these DNA probes (Table 2).

Changes in the cfDNA properties block the activity of another DNA sensor gene, *AIM2*. RNA*AIM2* and AIM2 protein levels are significantly reduced in the presence of sz-cfDNA and hc-cfDNA (Figure 4). We previously noted that changes in the activity of *AIM2* and *TLR9* in the presence of GC-rich and oxidized DNA negatively correlate in MCF7 cells [67]. This effect was confirmed for HSFs as well. The sz-cfDNA and hc-cfDNA (Figure 4F) caused the maximum changes in the RNA*TLR9*/RNA*AIM2* and TLR9/AIM2 ratios. After 24 h of HSFs being cultured in the presence of sz-cfDNA, there was a maximum decrease in the RNA *IL1B* level. The signaling pathway associated with the AIM2 protein regulates the level of this cytokine synthesis.

The gc-DNA, sz-cfDNA and hc-cfDNA probes significantly stimulate encoding STING protein *TMEM173* gene expression at both the transcriptional and translation level (Figure 5). cGAS-STING signaling plays a crucial role in inflammation [54,57,58]. Changes in the STING level correlate negatively with AIM2 level changes. For the sz-cfDNA sample, a positive correlation was found in changes in STING and TLR9 levels.

In half of the studied HSF lines, in the presence of DNA probes, an increase in activity at the level of transcription and translation of the DDX58 gene, which encodes the RIG-I protein, is observed. Changes in the RIG-I protein level correlate positively with changes in the levels of the AIM2 and EEA1 proteins and negatively with the changes in the levels of the STING and TLR9 proteins. TLR9 levels increase in HSFs in the presence of gc-DNA, sz-cfDNA, and hc-cfDNA probes is associated with decreases in RIG-I protein levels in the same HSF lines.

We also analyzed the changes in the *HMGB1* gene expression level, which encodes the HMGB1 protein. Changes in HMGB1 protein expression levels in the presence of sz-cfDNA and hc-cfDNA correlate with the changes observed in the presence of a gc-DNA probe at all periods of cultivation.

Thus, we have shown that sz-cfDNA and hc-cfDNA have the ability to change the activity of the DNA sensor genes *TLR9*, *STING*, *AIM2* and *RIG-I*, as well as the genes of the HMGB1 and EEA1 proteins, which are involved in the transport of DNA fragments into the cells. An increase in *TLR9* gene activity correlates with an increase in *STING* gene activity and occurs with blocking the activity of the genes *AIM2* and *RIG-I*. Sz-cfDNA and hc-cfDNA have the maximum similarity to the gc-DNA probe in terms of the effect on the studied DNA sensors. Sz-cfDNA, hc-cfDNA and gc-DNA stimulate the maximal cellular response on their presence in the culture medium.

Thus, the biological activity of sz-cfDNA and hc-cfDNA is apparently determined by the high GC-rich DNA fragment content in their composition. First, these fragments are easily oxidized, enhancing the ability of cfDNA to penetrate into the cytoplasm of cells; second, they contain a large number of binding motifs with TLR9. We believe that there is a mechanism regulating the interaction of *TLR9* and *AIM2*. An increase in GC-rich DNA fragments in cfDNA leads to an increase in *TLR9* expression and automatic blocking of *AIM2* expression. Further studies are required to investigate the details of this mechanism.

We found no significant differences in the biological activity of the sz-cfDNA and hc-cfDNA probes in HSFs. The sz-HSF and hc-HSF lines did not differ by the cellular response to the DNA probes. The significantly stronger HSF response to sz-cfDNA and hc-cfDNA probes compared to gDNA, oxy-DNA and gc-DNA probes is possibly explained by the combination of two sz-cfDNA and hc-cfDNA properties—high oxidation levels and a rich GC composition. In addition, the set of sequences altered in the composition of cfDNA composition compared to the cellular DNA [44], different methylation levels and various oxidative base modifications may be important. Some c-cfDNA fragments with a certain structure can more easily penetrate into the cell.

When planning the study, we expected to find biological activity differences between sz-cfDNA and hc-cfDNA introduced into the medium in the same amounts (50 ng/mL). We have previously found such differences for c-cfDNAs isolated from the plasma of patients with cardiovascular diseases and patients with rheumatoid arthritis [32,68]. We compared the characteristics of sz-cfDNA and other c-cfDNA samples studied previously. It turned out that the enrichment coefficient for the marker of GC-rich DNA (rDNA) for sz-cfDNA is only 1.4 times higher than for hc-cfDNA. C-cfDNA enrichment coefficient values for patients with rheumatoid arthritis were an order of magnitude higher than those for hc-cfDNA. Apparently, an identical sz-cfDNA and hc-cfDNA biological activity is explained by sz-cfDNA’s relatively low enrichment with GC-rich DNA fragments.

Despite the fact that in vitro sz-cfDNA and hc-cfDNA samples show approximately the same biological activity with regard to HSFs, the situation in the body may be different. The biological activity of c-cfDNA significantly depends not only on the content of the active fragments in the c-cfDNA pool but also on these fragments’ concentration in the intercellular environment. The concentration of GC-rich and oxidized fragments in the blood of schizophrenic patients is several folds higher than in the blood of healthy people [6,7]. An increase in the biologically active c-cfDNA concentration in the blood of schizophrenic patients can significantly increase its effect on the DNA sensors of various body cells, stimulating the inflammation that is characteristic of schizophrenia.

The limitation of the present study is produced by an in vitro cell model of cultured skin fibroblasts. Gene expression profiles and methylation patterns vary significantly between skin cells and brain cells. The response to the DNA fragments in the medium can also vary in the brain cells. At the same time, our recent studies have shown that the response of neurons to changes in cfDNA properties has much in common with the response of other tissues’ cells to cfDNA fragments [23,69,70]. In the brain, with an intact blood–brain barrier, cfDNA fragments circulate in the extracellular medium, the composition of which may differ significantly from the plasma cfDNA. However, it has been shown that changes in the plasma cfDNA composition are similar to the changes in the cerebrospinal fluid cfDNA composition—in both cases, an accumulation of GC-rich DNA repeats with high biological activity was found [71,72].

In addition, inflammation in schizophrenia is systemic. Cytokines and ROS, the synthesis of which stimulates cfDNA in body cells outside the brain, can affect the brain cells’ functioning, penetrating the barrier [73]. It is necessary to continue to investigate the role of cfDNA in mental illnesses in order to understand whether cfDNA may be a potential target of therapy.

We can outline two future directions for this research. The first one is a comparative analysis of the biological activity of individual plasma cfDNA samples from schizophrenia patients with various degrees of manifestation of the disease and with various levels of oxidative stress. As we have shown [6,7], the fraction of the active fragments in patients’ cfDNA varies significantly between patients. Secondly, we plan to conduct a study of the revealed bioactivity of schizophrenia patient cfDNA on brain and immune system cells.

## 4. Materials and Methods

### 4.1. Experimental Design

Five lines of HSFs obtained from healthy donors (hc-HSFs) and five lines obtained from SZ patients (sz-HSFs) were included in the study. The cfDNA concentration in the culture medium before the addition of DNA probes varied from 5 to 18 ng/mL (9 ± 5 ng/mL, n = 10). To simulate changes in the cfDNA properties, various DNA probes (50 ng/mL) were added to the cell culture medium for 1 h, 3 h and 24 h. A total of 180 HSFs were analyzed (10 HSF lines, 5 DNA probes and a control; 3 time intervals were used); each HSF was analyzed in triplicate.

### 4.2. Healthy Volunteers and Schizophrenia Patients

The study involved healthy volunteers (N = 10) and unmedicated schizophrenia patients (N = 10) aged 18 to 35 years (27 ± 8).

During the patients’ medical examination, clinical, psychometric, and pathopsychological methods adopted in psychiatric practice were used. The sample was homogeneous in terms of schizophrenia diagnosis, in accordance with the International Classification of Diseases of the tenth revision (ICD-10) criteria. The examination of the patients was carried out according to the standard of psychiatric anamnesis collection and mental status assessment; if necessary, objective data were applied (from medical documentation, from the words of relatives and third parties). The study of hereditary factors and the anamnesis of patients took into account the peculiarities of the course of pregnancy and childbirth in the mother, as well as information about the diseases she suffered.

The procedure of patients’ examination in the present study included:∗Analysis of sociodemographic characteristics;∗Characteristics of the course of the disease;∗Characteristics of the effect of drug therapy on the dynamics of the attack and the general course of the disease;∗Groups of symptoms established in ICD-10 and ICD-11 for the diagnosis of schizophrenia; groups of symptoms for the diagnosis of schizophrenia, as well as concomitant symptoms established in DSM-5;∗Clinical dimensions (dimensions) of the severity of psychotic symptoms (Clinical-Related Dimensions of Psychosis Symptom Severity);∗A scale of differentiated assessment of symptoms; assessment of the severity of clinical signs;∗Indicators of validated international psychometric scales (PANSS, BFCRS, NGS-A, etc.).

The examination was conducted in the conditions of round-the-clock inpatient units of the N. A. Alekseev Clinical Psychiatric Hospital No 1, where patients receive standard psychiatric care.

For patients with schizophrenia and for healthy controls, the following examinations were performed: urine analysis for psychotropic drugs to exclude psychotropic drug abusers; blood serology to exclude infectious diseases; expert-level EEG visual analysis and comparative EEG mapping to exclude any subclinical conditions.

Criteria for not being included in the study were as follows:(1)The presence of concomitant mental disorders, such as dependence on drugs and other psychoactive substances, organic mental disorders of any origin (except alcoholism), dementia, and mental retardation;(2)Severe somatic and chronic neurological diseases;(3)Severe acute and chronic somatic diseases preventing the examination, which has caused repeated hospitalizations, loss of work, led to disability, entailed the development of severe complications, such as stroke or heart attack, caused the development of acute or chronic insufficiency of internal organs or body systems, and which may affect the diagnosis, course of mental disorder, as well as the conducted drug therapy;(4)Refusal to cooperate during the implementation of research procedures.

### 4.3. Cell Culture

Primary adult human skin fibroblasts (HSFs) of healthy donors (n = 5) and SZ patients (n = 5) were obtained from the Research Centre for Medical Genetics collection. Before the treatments, HSFs were subcultured with 10% serum not more than 4 times. Fibroblasts were cultured in a medium supplemented with 10% serum until subconfluency. To analyze the response of cells to changes in the cfDNA properties, the cells were transferred to serum-free media “Hybris”, consisting of the basal medium and a serum-free supplement containing purified human albumin and growth factor cocktail (http://www.paneco.ru/, (accessed on 19 March 2022)) for 24 h. Various DNA samples (50 ng/mL) were added to the cell culture medium for 1 h, 3 h and 24 h to simulate changes in the cfDNA properties. A total of 180 HSF samples were analyzed (10 HSF lines, 5 DNA probes and a control; 3 time intervals). Each HSF sample was analyzed in triplicate.

### 4.4. Ethical Approval for the Use of Primary (Blood Leukocytes) and Cultured Human Cells

The study was carried out in accordance with the latest version of the Declaration of Helsinki and was approved by the Independent Interdisciplinary Ethics Committee on Ethical Review for Clinical Studies (Protocol №4 (dated 15 March 2019) for the scientific minimally interventional study “Molecular and neurophysiological markers of endogenous human psychoses”). All participants signed an informed written consent to participate in the study after the procedures had been completely explained.

### 4.5. DNA Probes

*gDNA*, *hc-cfDNA*: DNA was isolated from the blood leukocytes and plasma of 10 healthy male donors (21–35 years old). The GDNA probe was isolated from blood leukocytes of healthy donors and digested with DNase 1 to obtain 15–10 kb fragments. This DNA probe contained relatively few rDNA copies. DNA oxidation (8-oxodG marker) was also low. *Oxy-gDNA*: gDNA was oxidized with hydrogen peroxide as described previously [74]. *sz-cfDNA*: DNA was isolated from the blood plasma of 10 untreated male SZ patients (age 18–29 y.o., without other diseases). DNA from the plasma and blood leukocytes was isolated by extraction with organic solvents, and the rDNA content in the cfDNAs was determined by the non-radioactive quantitative hybridization. The techniques have been described in detail previously and have been applied without modification [75].

The content of the oxidation marker 8-oxodG in cfDNA and gDNA was determined by enzyme immunoassay as described previously [6]. Genomic DNA with the known 8-oxodG amount was used in order to plot a calibration curve of the signal intensity dependence on the 8-oxodG content in the sample. The 8-oxodG content in the control oxy-gDNA samples was determined by ESI-MS/MS method using an AB SCIEX 3200 Qtrap machine [74].

*gc-DNA**:* linearized plasmid DNA (10 197 bp) contains rDNA sequences (5836 bp, 73% GC) cloned into the EcoRI site of the pBR322 vector 4361 bp (53% GC) in length. A cloned rDNA fragment covers positions from −515 to 5321 of human rDNA according to HSU13369, GeneBank. The linearized vector pBR322 served as a control.

Table 3 presents a summarized description of the five DNA probes used.

All DNA samples were subjected to the same lipopolysaccharide-removing purification procedure that included treatment with Triton X-114 followed by gel filtration on the HW-85.

Synthesis of fluorescently labeled DNA probes. To obtain FL-DNA probes, we used a set of reagents, including the Label IT Nucleic Acid Labeling Kits, Fluorescein (MIR 3200), Rhodamine (MIR 4100), (Mirus Bio LLC, St. Louis, MO, USA), which directly interact with chemical groups of DNA.

### 4.6. Flow Cytometry Analysis (FCA)

To analyze the localization of cfDNA fragments in HSFs, fluorescently labeled DNA probes (50 ng/mL) were added to the cell culture medium for 1 h. Each probe showed approximately the same pattern (Figure 1B). The labels were located mainly in the cytoplasm.

To determine whether DNA is located on the surface of the cells or penetrates across the membrane, we used FCA. Treatment of cells with EDTA and trypsin solutions leads to the desorption of cfDNA fragments from the cell surface [76,77].

HSFs were grown for 1 h, 3 h or 24 h in 60 mm dishes either in the absence or presence of various DNA probes. Before FCA, the cells were washed in Versene solution, then treated with 0.25% trypsin under light microscope observation. Cells were transferred to the Eppendorf tubes, washed with culture media and then centrifuged and resuspended in PBS. Staining the cells with antibodies was performed. Briefly, to fix the cells, paraformaldehyde (Sigma, Kawasaki, Japan) was added at a final concentration of 3% at 37 °C for 10 min. Then the cells were washed 3 times with 0.5% BSA-PBS and permeabilized with 0.1% Triton X-100 (Sigma) in PBS. Cells (~50 × 10^3^) were washed 3 times with 0.5% BSA-PBS and stained with primary antibodies (working dilution 1 µg/mL) to EEA1 (ab2900 Abcam, Bristol, UK), TLR9 (sc-515921 Santa Cruz Biotechnology, Inc., Dallas, TX, USA), and STING (BioLegend, San Diego, CA, USA), RIG-I (BioLegend USA) overnight at 4 C. Thy were then washed twice and incubated 1h in the dark with the following secondary antibodies (working dilution 0.5 µg/mL): m-IgGk BP-FITC (sc-516140), m-IgGk BP-PE (sc-516141), anti-rabbit IgG-PE (sc-3753) (Santa Cruz Biotechnology, Inc., USA). They were then again washed thrice with 0.5% BSA-PBS. When stained with the conjugated antibodies FITC-HMGB1 (NB100-2322F Novus Biologicals, LLC, Centennial, CO, USA) and AF350-AIM2 (bs-5986r-a350 Bioss Inc., Woburn, MA, USA), cells were incubated for 3 h at 4 °C in the dark at room temperature and washed thrice with 0.5% BSA-PBS.

To quantify the background fluorescence, we stained a portion of the cells with secondary FITC(PE)-conjugated antibodies only. Cells were analyzed at CytoFLEX S (Beckman Coulter, Brea, CA, USA). Primary data are presented as median signal values minus signal background values. The relative standard error of the FCA was 4 ± 2%.

### 4.7. Real-Time PCR Assay

RNA was extracted from the cells using YellowSolve kits (Klonogen, St.-Petersburg, Russia) or Trizol reagent (Invitrogen, Carlsbad, CA, USA) according to the specified method (http://tools.lifetechnologies.com/content/sfs/manuals/trizol_reagent.pdf, accessed on 22 March 2021). Phenol-chloroform extraction and precipitation with chloroform and isoamyl alcohol (49:1) were performed. The RNA concentration was determined using a Quant-iT RiboGreen RNA reagent dye (R11491 Invitrogen, Carlsbad, CA, USA) on a plate reader EnSpire (PerkinElmer, Waltham, MA, USA), em = 487 nm, fl = 524 nm. According to the standard protocol, the reverse transcription reaction was performed using reagents from Sileks (Moscow, Russia). PCR was performed using the specific primers and Sybr-Green intercalating dye on a StepOnePlus device (Applied Biosystems, Foster City, CA, USA). The selection and synthesis of the primers were performed by Evrogen (Moskow, Russia).

The reaction PCR mixture in a volume of 25 µL consisted of 2.5µL PCR buffer (700 mmol/L Tris-HCl, pH 8.6; 166 mmol/L ammonium sulfate, 35 mmol/L MgCl2), 2 µL 1.5 mmol/L dNTP solution; and 1 µL 30 pmol/L primer solution, cDNA. PCR conditions were selected individually for each primer pair. After denaturation for 4 min at 95 °C, 40 amplification cycles were performed in the following order: 94 °C for 20 s, 56–62 °C for 30 s, 72 °C for 30 s, and 72 °C for 5 min. The results were processed using a calibration plot. The error was 2%.

### 4.8. Fluorescence Microscopy

An Axio Scope.A1 microscope (Carl Zeiss, Oberkochen, Germany) was used for fluorescent microscopy of the cells.

Immunocytochemistry. HSFs were fixed in 3% formaldehyde (4 °C) for 20 min, washed with PBS and then permeabilized with 0.1% Triton X-100 in PBS for 15 min at room temperature. This was followed by blocking with 0.5% BSA in PBS for 1 h and incubating overnight at 4 °C with the PE-TLR9, PE-AIM2, FITC-HMGB1 antibodies. After washing with 0.01% Triton X-100 in PBS, HSFs were washed with PBS and then stained with 2 μg/mL DAPI.

Intracelullar localization of labeled DNA probes. (1) Unfixed cells. DNA probe–Red (rhodamine) and DNA–probe–Green (fluorescein) were added to cultivation media for 1 h (50 ng/mL). Cells were washed three times with PBS. (2) DNA probes were added to cultivation media for 1h (50 ng/mL). Cells were washed three times with PBS, fixed in 3% paraformaldehyde (4 °C) for 20 min, washed with PBS and stained with PE-TLR9 or FITC-HMGB1. After washing with 0.01% Triton X-100 in PBS, HSFs were washed with PBS and then stained with 2 μg/mL DAPI.

### 4.9. Statistical Analysis

Experiments were repeated in triplicate. In FCA, the medians of the signal intensities were analyzed. The significance of the observed differences was analyzed with the nonparametric Mann–Whitney U test. The *p*-values < 0.01 were considered statistically significant. The data were analyzed with Excel, Microsoft Office (Microsoft, Redmond, WA, USA), StatPlus2007 Professional software (http://www.analystsoft.com (accessed on 19 March 2022) and StatGraphics (Statgraphics Technologies, The Plains, VA, USA).

## Figures and Tables

**Figure 1 genes-13-00551-f001:**
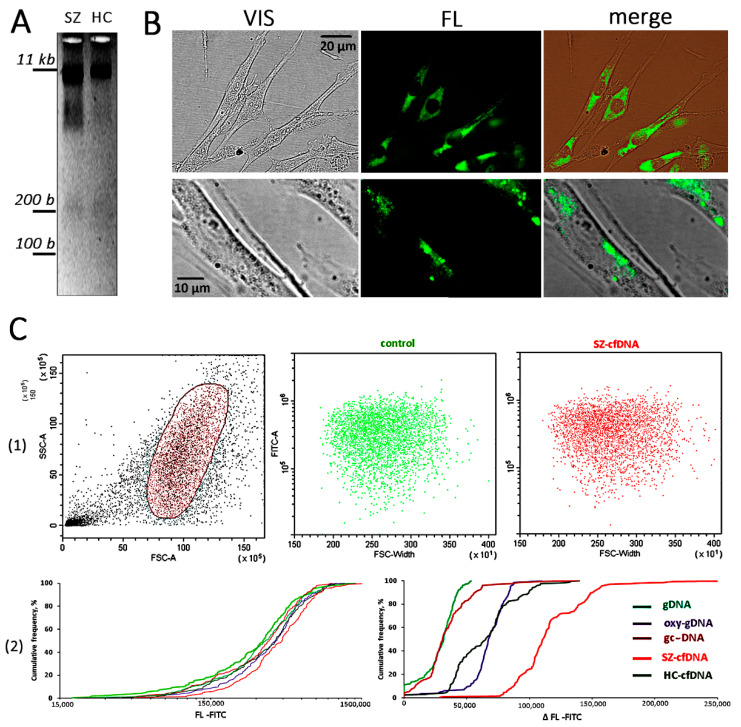
Interaction of DNA probes with HSFs. (**A**) Electrophoresis of sz-cfDNA and hc-cfDNA in 1.2% agarose gel stained with ethidium bromide. (**B**) Visualization of fluorescently (Fluorescein) labeled DNA probes in HSFs. An example is given for the sz-cfDNA sample and the sz-HSF1 line. (**C**) Analysis of fluorescently (Fluorescein) labeled DNA probes in HSFs using FCA. (1) Data from the device for sz-HSF1 are given. (2) Cumulative distributions of the sz-HSF1 cells according to the FL-signals. Curves for different DNA probes are shown in different colors. The cells were incubated for 1 h in the presence of the DNA probes (50 ng/mL).

**Figure 2 genes-13-00551-f002:**
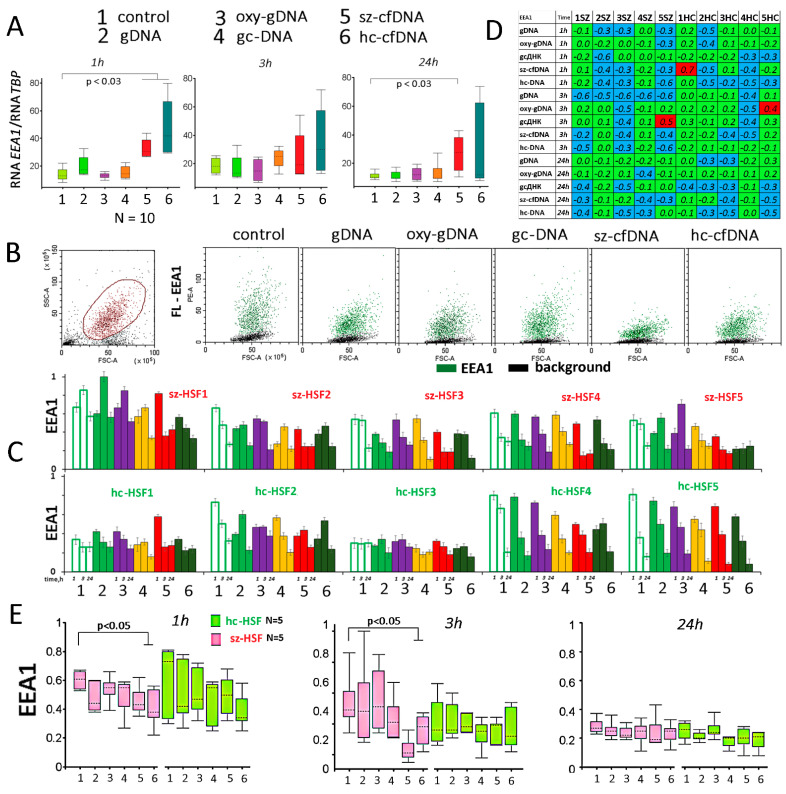
The influence of the DNA probes on the expression of *EEA1* gene in HSFs. (**A**) Change of the RNA*EEA1* in HSFs (n = 10) in the presence of DNA probes (50 ng/mL). The cultivation time and the DNA probes are indicated on the graph. Sz-HSFs did not differ from hc-HSFs in terms of the RNA*EEA1* levels. (**B**) The most typical examples of the EEA1 analysis with FCA in HSFs. The data from the device are given for sz-HSF4 (24 h). EEA1: green color; background: black color. (**C**) Index EEA1: the values of the medians of FL-EEA1, normalized to the maximum signal value in the sample (n = 180). Average values for three measurements and standard deviation are given. (**D**) Analysis of relative changes in EEA1 levels in the presence of DNA probes compared to control. Figures indicate the ratio (EEA1probe DNA—EEA1control)/EEA1control. Green color—no significant differences with the control (*p* > 0.01), red—increased protein content in the presence of the DNA probe (*p* < 0.01) and blue—decreased protein content (*p* < 0.01). (**E**) Changes in the protein EEA1 in samples of sz-HSFs (n = 5) and hc-HSFs (n = 5).

**Figure 3 genes-13-00551-f003:**
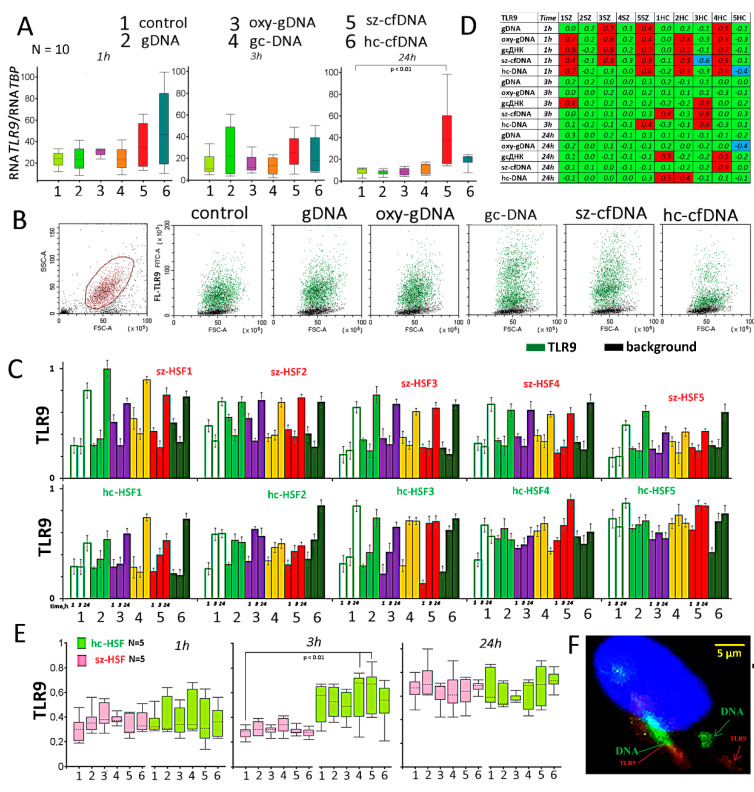
The influence of the DNA probes on the expression of *TLR9* gene in HSFs. (**A**) Change of TLR9 in HSFs (n = 10) in the presence of DNA probes (50 ng/mL). The cultivation time and the DNA probes are indicated on the graph. Sz-HSFs did not differ from hc-HSFs in terms of the RNA*TLR9* levels. (**B**) The most typical examples of the TLR9 analysis with FCA in HSFs (FCA). The data from the device are given for sz-HSF4 (24 h). TLR9: green color; background: black color. (**C**) Index TLR9: the values of the medians of FL-TLR9, normalized to the maximum signal value in the sample (n = 180). Average values for three measurements and standard deviation are given. (**D**) Analysis of relative changes in TLR9 levels in the presence of DNA probes compared to the control. Figures indicate the ratio (TLR9 probe DNA—TLR9 control)/TLR9 control. Green color—no significant differences with the control (*p* > 0.01), red—increased protein content in the presence of the DNA probe (*p* < 0.01) and blue—decreased protein content (*p* < 0.01). Changes in the protein TLR9 in samples of sz-HSFs (n = 5) and hc-HSFs (n = 5). (**E**) Visualization of labeled DNA probes (green) and TLR9 (red) in HSFs. An example is given for the gc-DNA probe and the sz-HSF1 (1 h).

**Figure 4 genes-13-00551-f004:**
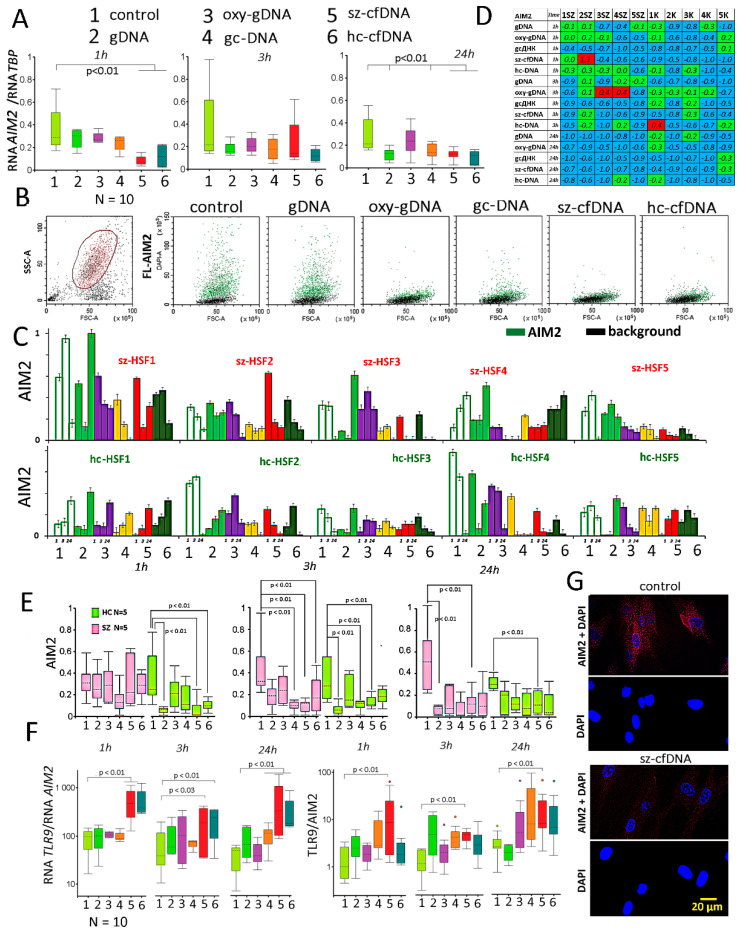
The influence of the DNA probes on the expression of *AIM2* gene in HSFs. (**A**) Change in the RNA*AIM2* in HSFs (n = 10) in the presence of DNA probes (50 ng/mL. The cultivation time and the DNA probes are indicated on the graph. Sz-HSFs did not differ from hc-HSFs in terms of the RNA*AIM2* levels. (**B**) The most typical examples of the AIM2 analysis with FCA in HSFs. The data from the device are given for hc-HSF5 (1 h). EEA1: green color; background: black color. (**C**) Index AIM2: the values of the medians of FL-AIM2, normalized to the maximum signal value in the sample (n = 180). Average values for three measurements and standard deviation are given. (**D**) Analysis of relative changes in AIM2 levels in the presence of DNA probes compared to the control. Figures indicate the ratio (AIM2 probe DNA—AIM2 control)/AIM2 control. Green color—no significant differences with the control (*p* > 0.01), red—increased protein content in the presence of the DNA probe (*p* < 0.01) and blue—decreased protein content (*p* < 0.01). (**E**) Changes in AIM2 level in hc-HSFs (n = 5). (**F**) Changes in the ratios RNA *TLR9* to RNA *AIM2* and TLR9 to AIM2 in samples of HSFs (n = 10). (**G**) Visualization of AIM2 (PE) in HSFs. An example is given for the sz-cfDNA probe and the sz-HSF1 (1 h).

**Figure 5 genes-13-00551-f005:**
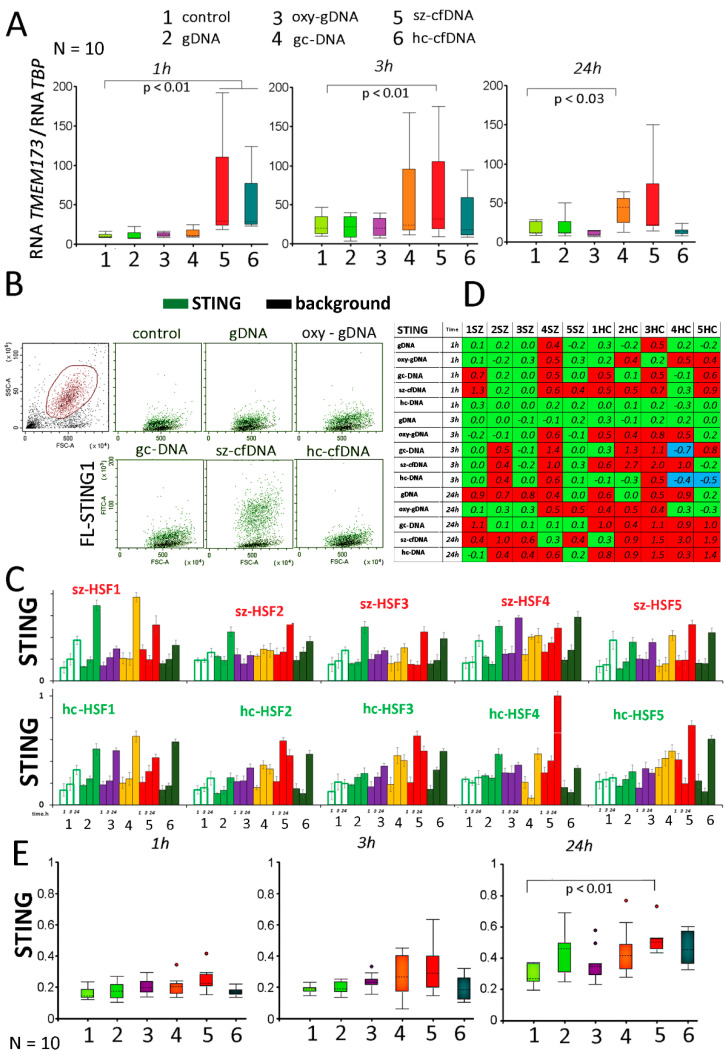
The influence of DNA probes on the expression of *TMEM173* gene in HSFs. (**A**) Change in the RNA *TMEM173* in HSFs (n = 10) in the presence of DNA probes (50 ng/mL). The cultivation time and the DNA probes are indicated on the graph. Sz-HSFs did not differ from hc-HSFs in terms of the RNA *TMEM173* levels. (**B**) The most typical examples of the STING analysis in HSFs (FCA). The data from the device are given for hc-HSF4 (24 h). STING: green color; background: black color. (**C**) Index STING: the values of the medians of FL- STING, normalized to the maximum signal value in the sample (n = 180). Average values for three measurements and standard deviation are given. (**D**) Analysis of relative changes in STING levels in the presence of DNA probes compared to the control. Figures indicate the ratio (STING probe DNA—STING control)/STING control). Green color—no significant differences with the control (*p* > 0.01), red—increased protein content in the presence of the DNA probe (*p* < 0.01) and blue—decreased protein content (*p* < 0.01). (**E**) Changes in the protein STING in samples of HSFs (n = 10). Sz-HSFs did not differ from hc-HSFs in terms of the STING levels (*p* > 0.05).

**Figure 6 genes-13-00551-f006:**
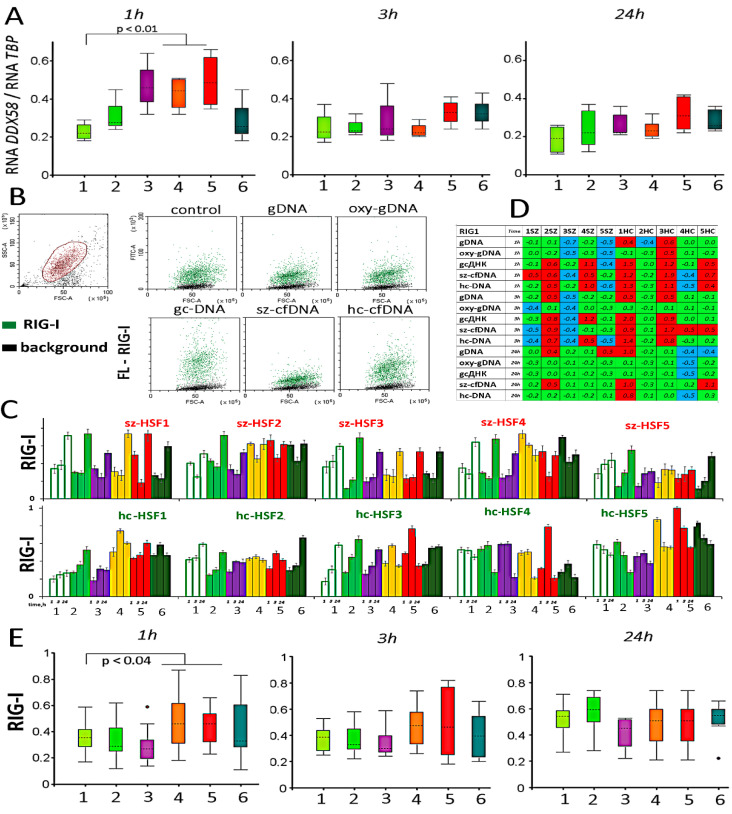
The influence of DNA probes on the expression of the *DDX58* gene in HSFs. (**A**) Change of the RNA *DDX58* in HSFs (n = 10) in the presence of DNA probes (50 ng/mL). The cultivation time and the DNA probes are indicated on the graph. Sz-HSFs did not differ from hc-HSFs in terms of the RNA *DDX58* levels. (**B**) The most typical examples of the RIG-I analysis with FCA in HSFs. The data from the device are given for hc-HSF1 (1 h). RIG-I: green color; background: black color. (**C**) Index RIG-I: the values of the medians of FL—RIG-I, normalized to the maximum signal value in the sample (n = 180). Average values for three measurements and standard deviation are given. (**D**) Analysis of relative changes in RIG-I levels in the presence of DNA probes compared to the control. Figures indicate the ratio (RIG-I probe DNA—RIG-I control)/RIG-I control). Green color—no significant differences with the control (*p* > 0.01), red—increased protein content in the presence of the DNA probe (*p* < 0.01) and blue—decreased protein content (*p* < 0.01). (**E**) Changes in the protein RIG-I in samples of HSFs (n = 10). Sz-HSFs did not differ from hc-HSFs in terms of the RIG-I levels (*p* > 0.01).

**Figure 7 genes-13-00551-f007:**
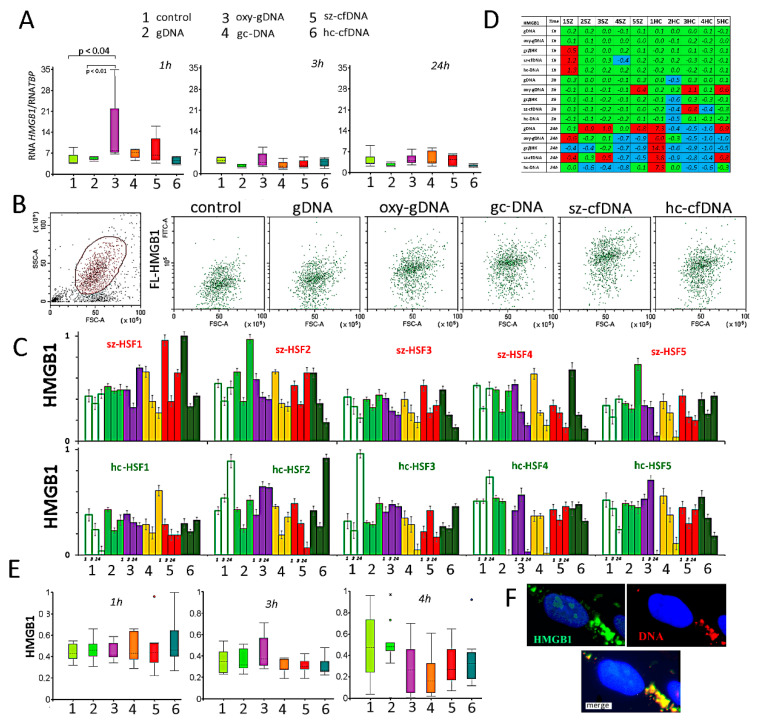
The influence of DNA probes on the expression of the *HMGB1* gene in HSFs. (**A**) Change of the RNA *HMGB1* in HSFs (n = 10) in the presence of DNA probes (50 ng/mL). The cultivation time and the DNA probes are indicated on the graph. Sz-HSFs did not differ from hc-HSFs in terms of the RNA *HMGB1* levels. (**B**) The most typical examples of the HMGB1 analysis with FCA in HSFs. The data from the device are given for sz-HSF3(1 h). HMGB1: green color; background: black color. (**C**) Index RIG-I: the values of the medians of FL-HMGB1, normalized to the maximum signal value in the sample (n = 180). Average values for three measurements and standard deviation are given. (**D**) Analysis of relative changes in HMGB1 levels in the presence of DNA probes compared to the control. Figures indicate the ratio (HMGB1 probe DNA—HMGB1control)/HMGB1 control). Green color—no significant differences with the control (*p* > 0.01), red—increased protein content in the presence of the DNA probe (*p* < 0.01) and blue—decreased protein content (*p* < 0.01). (**E**) Changes in the protein HMGB1 in samples of HSFs (n = 10). Sz-HSFs did not differ from hc-HSFs in terms of the HMGB1 levels (*p* > 0.01). (**F**) Localization of the labeled DNA probes (red) and HMGB1 (green) in HSFs. An example is given for the sz-cfDNA (1 h) sample and the sz-HSF1 line.

**Table 1 genes-13-00551-t001:** Spearman’s rank correlation (Rs and *p*-value) for the protein levels in HSFs incubated with DNA probes.

gDNA	TLR9	RIG-I	1 h	−0.67	0.035
	TLR9	HMGB1	3 h	0.66	0.037
	AIM2	HMGB1	24 h	0.70	0.024
	RIG-I	HMGB1	1 h	0.65	0.042
	RIG-I	HMGB1	24 h	0.76	0.011
	RIG-I	EEA1	1 h	0.64	0.044
oxy-DNA	TLR9	STING	24 h	0.65	0.041
	AIM2	HMGB1	24 h	0.71	0.022
	AIM2	STING	1 h	−0.65	0.042
	HMGB1	STING	1 h	−0.91	0.0002
gc-DNA	TLR9	RIG-I	1 h	−0.72	0.018
	TLR9	HMGB1	3 h	0.90	0.0004
	AIM2	HMGB1	3 h	0.70	0.024
	AIM2	RIG-I	3 h	0.69	0.027
	STING	RIG-I	24 h	−0.65	0.042
sz-cfDNA	TLR9	RIG-I	1 h	−0.73	0.017
	TLR9	HMGB1	3 h	0.79	0.007
	TLR9	STING	24 h	0.80	0.006
	AIM2	RIG-I	3 h	0.93	0.0001
	AIM2	HMGB1	3 h	0.85	0.002
	AIM2	HMGB1	24 h	0.67	0.033
	RIG-I	HMGB1	3 h	0.66	0.037
	RIG-I	EEA1	1 h	0.64	0.047
hc-cfDNA	TLR9	RIG-I	1 h	−0.75	0.012
	TLR9	HMGB1	3 h	0.75	0.012
	AIM2	RIG-I	3 h	0.90	0.0004
	AIM2	RIG-I	1 h	0.70	0.025
	RIG-I	EEA1	1 h	0.74	0.014

**Table 2 genes-13-00551-t002:** Analysis of the relative changes in the average RNA levels in the presence of DNA probes compared to the controls for ten HSFs. Figures indicate the ratio: median (RNAi probe DNA/RNAi control). Green color—no significant differences with the control (*p* > 0.01); yellow color—increased RNA content in the presence of the DNA probe (*p* < 0.01); red color—maximum increase in RNA content in the presence of the DNA probe and blue—decreased RNA content (*p* < 0.001).

#	*Genes*	1 h					3 h					24 h				
		gDNA	oxy-DNA	gc-DNA	sz-cfDNA	hc-cfDNA	gDNA	oxy-DNA	gc-DNA	sz-cfDNA	hc-cfDNA	gDNA	oxy-DNA	gc-DNA	sz-cfDNA	hc-cfDNA
1	*NFKB1*	1.5	1.3	1.9	2.8	1.6	0.4	0.9	0.6	2.2	1.6	0.6	1.1	1.6	1.2	1.1
2	*STAT3*	2.5	5.8	2.0	5.4	8.7	0.8	0.9	1.4	2.6	3.4	0.7	1.8	2.7	2.8	9.8
3	*STAT6*	0.7	0.6	0.7	2.3	3.2	0.6	0.5	0.8	1.4	1.2	0.3	0.6	0.6	1.9	3.6
4	*PPARG2*	1.5	0.9	1.3	2.2	1.8	1.0	0.8	2.4	2.2	0.4	1.5	0.8	2.6	0.8	1.1
5	*MTOR*	2.2	1.0	2.0	7.9	11.7	0.9	0.7	1.3	3.0	4.2	0.5	0.6	1.4	1.4	4.7
6	*AKT2*	2.5	0.8	2.6	8.7	5.9	1.0	1.6	0.7	1.0	2.0	1.3	1.6	2.6	2.4	5.3
7	*MAPK1*	0.9	0.6	0.8	1.6	2.5	0.8	0.6	0.6	0.8	1.3	0.4	0.5	0.9	3.7	1.6
8	*MAPK3*	2.0	2.6	2.3	3.9	2.4	0.9	1.5	0.6	1.7	0.9	1.1	1.8	3.1	1.0	1.3
9	*TP53*	0.8	0.7	1.0	4.0	6.9	0.8	3.6	1.4	0.8	1.7	0.9	0.8	2.9	2.5	9.7
10	*ATM*	1.5	1.6	2.2	11.3	15.6	1.2	0.9	1.1	2.4	1.4	0.5	1.1	1.4	1.8	14.1
11	*ATR*	0.6	0.8	1.5	1.5	3.3	0.9	0.8	1.1	10.0	3.9	0.6	1.6	1.7	8.8	16.3
12	*APAF1*	0.9	0.9	1.0	3.6	4.8	0.7	1.7	1.1	1.2	1.8	0.4	0.7	0.7	0.5	4.1
13	*AIFM1*	1.7	1.2	1.9	12.3	16.0	0.7	0.8	0.8	1.9	2.4	0.5	1.3	1.7	1.5	7.5
14	*BAX*	4.4	1.4	6.3	8.6	3.1	0.8	2.8	0.7	0.7	1.0	0.9	1.4	2.7	0.3	1.4
15	*BAK1*	1.4	1.2	1.7	11.9	19.0	0.9	0.8	1.2	2.4	2.8	0.5	1.3	2.0	20.9	18.3
16	*BIRC2*	1.5	1.0	1.5	2.8	4.0	0.8	1.4	0.7	0.6	0.3	0.4	0.8	0.9	0.7	2.0
17	*BIRC3*	0.9	0.8	1.1	2.0	3.2	0.7	0.9	0.6	1.5	1.1	0.5	0.7	1.6	1.8	6.7
18	*BCL2*	2.3	1.2	2.1	20.3	18.4	0.5	1.3	1.2	2.3	2.3	0.9	1.4	3.2	3.3	12.0
19	*BCL2A1*	1.6	1.2	1.5	2.1	3.1	1.3	1.6	2.7	2.5	3.8	0.8	1.6	3.7	1.9	14.0
20	*BCL2L1*	4.5	1.2	6.6	10.4	4.2	0.8	3.0	0.7	0.6	1.2	1.0	1.7	2.8	0.5	2.5
21	*CCND1*	3.0	2.6	2.9	9.3	9.0	1.0	1.3	1.6	1.2	2.5	6.4	1.4	3.4	1.2	6.9
22	*CDKN2A*	1.6	1.4	3.0	3.8	2.9	0.9	2.9	0.8	0.3	0.9	0.4	0.6	1.4	0.7	2.1
23	*CDKN1A*	2.5	1.3	3.5	4.6	1.7	0.3	3.0	0.7	0.8	0.9	0.9	1.3	2.9	0.2	1.1
24	*ICAM*	1.4	0.9	1.3	0.7	0.6	0.4	0.9	0.5	0.5	0.5	0.3	0.7	0.9	0.1	0.4
25	*VCAM*	1.3	1.0	1.4	1.7	3.4	0.4	1.3	0.8	0.7	0.4	0.5	0.8	1.3	0.6	14.5
26	*SELE*	1.4	0.9	2.7	9.7	13.4	0.7	1.2	1.4	1.8	2.0	0.4	0.6	1.2	3.2	3.7
27	*MFN1*	1.7	0.7	1.2	8.7	8.8	1.0	0.9	2.8	6.5	1.7	1.2	1.7	3.1	3.3	4.8
28	*FIS1*	3.7	1.1	6.3	9.5	3.9	0.3	3.0	0.8	0.7	3.9	1.5	2.2	2.4	1.1	1.1
29	*ENDOG*	1.7	1.3	1.1	4.4	5.9	0.9	0.9	0.7	2.0	1.8	0.6	0.9	1.9	2.6	6.7
30	*ATG16L1*	1.5	1.0	1.2	8.2	13.4	0.9	0.9	1.7	2.6	3.8	0.6	1.2	2.1	5.6	7.6
31	*BECN1*	4.2	2.0	4.5	13.0	6.7	0.4	1.8	1.5	1.2	2.6	2.0	1.9	14.7	5.4	10.1
32	*IL-8*	1.6	1.1	6.3	7.7	7.5	0.9	1.4	0.7	1.0	0.9	0.3	0.8	1.2	1.1	1.5
33	*IL1B*	1.1	0.9	1.3	3.0	6.6	0.4	1.6	0.5	0.8	0.6	0.5	1.1	2.2	0.4	1.8

**Table 3 genes-13-00551-t003:** Characteristics of the DNA probes.

DNA Probe	Origin of the DNA Sample	rDNA CN (%)	8-oxodG/10^6^ N	Length, kb
gDNA	HC leukocytes (n = 10)	420 ± 48	0.5 ± 0.1	10–15
oxy-gDNA	gDNA/H2O2	401 ± 59	110 ± 49	7–11 *
gc-DNA	pBR322-rDNA	70%	11 ± 10	10.2
sz-cfDNA	SZ plasma (n = 10)	1060 ± 75	65 ± 21	6–10 *
hc-cfDNA	HC plasma (n = 10)	756 ± 36	24 ± 17	8–10 *

* Samples contain low-molecular-weight DNA fragments (Figure 1A).

## Data Availability

The data presented in this study are available on request from the corresponding author. The data are not publicly available due to patient confidentiality.

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
