# Peer review of "In Vitro Analysis of Biological Activity of Circulating Cell-Free DNA Isolated from Blood Plasma of Schizophrenic Patients and Healthy Controls"

_genes, 2022, doi:10.3390/genes13030551_

Round 1

Reviewer 1 Report

REVIEWER’S COMMENTS

The manuscript In Vitro Analysis of Biological Activity of Circulating Cell-Free DNA Isolated from Blood Plasma of Schizophrenic Patients and Healthy Controlsby Ershova et al investigated in vitro response of ten human skin fibroblast lines to five DNA probes containing different amounts of a GC-rich marker and a DNA-oxidation marker, including sz-cfDNA and healthy control c-cfDNA (hc-cfDNA) probes.

  1. Scale bars are missing from Figures 1B, 3F, 4G, and 7F.
  2. Please improve the resolution of figures. The resolution should be at least 300 dpi.
  3. In the methods section, please include catalog numbers and manufacturer of assay kits, primers, and antibodies. Please include the dilution/ concentration of primary and secondary antibodies used in this study.
  4. Please provide a reference for the following mentioned in the last paragraph of the discussion – “inflammation in schizophrenia is systemic. Cytokines and ROS, the synthesis of which stimulates cfDNA in body cells outside the brain, can affect the brain cells functioning penetrating the barrier.”
  5. Please discuss briefly the future directions for this research.
  6. Please be consistent with the style of references.

Author Response

We thank the reviewer for the very useful notes. Below are point-by-point replies for the reviewer's comments:
1. We added Scale bars to the Figures
2. The resolutions was initially 762 dpi (300 dots per cm). Apparently, the resolution was reduced in the figures embedded to the Word file. We attached the figures as separate files with resolution 762 dpi
3. The catalog numbers are included in the revised Manuscript
4. The reference has been added (#78)
5. We briefly outlined the future directions. Please, see the last paragraph of Discussion in the revised version
6. Thank you for your attention to the small details. We corrected the reference style

Reviewer 2 Report

Reviewer: NM

Review paper Genes (ISSN 2073-4425)

Title : " In Vitro Analysis of Biological Activity of Circulating Cell-Free DNA Isolated from Blood Plasma of Schizophrenic Patients and Healthy Controls"

The paper # genes-1614132 titled " In Vitro Analysis of Biological Activity of Circulating Cell-Free DNA Isolated from Blood Plasma of Schizophrenic Patients and Healthy Controls" byElizaveta S. Ershova , Galina V. Shmarina , Lev N. Porokhovnik * , Natalia V. Zakharova , George P. Kostyuk , Pavel E. Umriukhin , Sergey I. Kutsev , Vasilina A. Sergeeva , Natalia N. Veiko , Svetlana V. Kostyuk, comparing the biological activity of the ccfDNA samples isolated from blood plasma of healthy controls and SZ patients, on human skin fibroblasts (HSFs) as an in vitro model. The authors describe the changes in genes expression of nucleic acid sensor proteins (TLR9,AIM2, STING, RIG-I) and proteins involved in the biopolymers transfection into cells (EEA1 and HMGB1) at transcriptional and translational levels in human cells in response to cfDNA fragments with different properties, including sz-cfDNA, hc-cfDNA and the model gDNA, oxy-DNA and gc-DNA

In particular,

  1. the abstract and the introduction should be implement with the explanation of the pathogenetic involvement of DMAP system and its role in the interplay between circulating cell-free DNA (c-cfDNA) and innate immunity. These concepts are partially reported in the discussion.
  2. the experimental procedure focuses on the biological effects induced by cf-DNA on secondary skin cell lines. In the blood the hematological cells and in particular the immune cells are those cell elements most involved in the interaction with circulating molecular products. On this way my question is whether it is appropriate to add an experimental set concerning direct effect on immune cells like macrophages and /or lymphocytes

  1. moreover, regarding the major research question was: can c-cfDNA of schizophrenic patients (sz-cfDNA) stimulate the DNA sensors genes, which control the innate immunity? Should be interesting understand as heat shock proteins (hsp) are involved in this potential relationship. The HSP or stress proteins, are highly conserved and present in all organisms and in all cells of all organisms. Selected HSPs, also known as chaperones, play crucial roles in folding/unfolding of proteins, assembly of multiprotein complexes, transport/sorting of proteins into correct subcellular compartments, cell-cycle control and signaling, and protection of cells against stress/apoptosis. More recently, HSPs have been implicated in antigen presentation with the role of chaperoning and transferring antigenic peptides to the class I and class II molecules of the major histocompatibility complex, absolving a crucial role in the biomodulation of innate immunity responsivity to unspecific stimulation, like those c-cfDNA induced

minor issues

- the English language should be improved

Author Response

We highly appreciate the Reviewer's valuable comments and are pleased to provide a point-by-point response below:
1. We're very flattered at your interest to the topic of DAMP signalling. However, the Abstract is already 270 words long (while ~200 is the recommended length), the Introduction is long enough, and the reference list contains 80 references. Further enlargement of the manuscript with backgroung fact description/explanation would turn the manuscript into a review from the experimental report. Our opinion is that an interested reader could easily refer the relevant cited references, in particular, ##6,7,30,68,73,78 in order to get more information on the subject. Most those references are open-access.
2. We dispose some data on the influence of schizophrenia patient cfDNA upon lymphocytes isolated from blood of patients and healthy controls. In general, the lymphocytes responded to the exposure to cfDNA by activation of expression of all the genes studied in the manuscript. However, some difference existed.
Interestingly, healthy control lymphocytes showed a much stronger response to the exposure to various cfDNA samples than patient lymphocytes. We project to publish these data separately. The aim of this pilot study described in the Manuscript was revealing the fact of cfDNA bioactivity itself.
We added a paragraph in the end of Discussion section in the revised version, where we outlined the possible future directions of our studies.
3. We greatly appreciate  this Reviewer's suggestion. We'll certainly conduct the analysis of changes in the gene expression profiles of heat shock proteins in response to the exposure to cfDNA and publish the report in the nearest future.

Round 2

Reviewer 2 Report

Dear Editor,
the authors' answers are interesting but, really, they did not accept any of the suggestions given. THE TEXT IS EXACTLY THE SAME as before my review. I therefore refer the decision on the publication of the manuscript to the editor
Best regards 

Author Response

Dear Reviewer and Dear Editor,

This is a technical error. Indeed, we changed the text. And, as recomended by the Editorial Office's guidelines, we made changes in the MS Word's change tracking mode. Please see attached some screenshots from a desktop PC monitor showing how we can see it. The attached file does not comprise ALL the changes we made, but just several examples.
Two hypothetical causes can be suggested to explain why the Reviewer saw "exactly the same text":
1) the Reviewer's computer failed to display changes due to wrong Word settings at his/her computer;
2) because of a technical error, the Reviewer received the previous version of the file.
So, we'll send the revised file one more time and ask to kindly analize again carefully.
Best regards,
The Authors

Round 3

Reviewer 2 Report

accept after minor revision on text editing